# Stochastic Quantized Activation: to prevent Overfitting in Fast Adversarial Training

## Abstract

Existing neural networks are vulnerable to "adversarial examples"—created by adding maliciously designed small perturbations in inputs to induce a misclassification by the networks. The most investigated defense strategy is adversarial training which augments training data with adversarial examples. However, applying single-step adversaries in adversarial training does not support the robustness of the networks, instead, they will even make the networks to be overfitted. In contrast to the single-step, multi-step training results in the state-of-the-art performance on MNIST and CIFAR10, yet it needs a massive amount of time. Therefore, we propose a method, Stochastic Quantized Activation (SQA) that solves overfitting problems in single-step adversarial training and fastly achieves the robustness comparable to the multi-step. SQA attenuates the adversarial effects by providing random selectivity to activation functions and allows the network to learn robustness with only single-step training. Throughout the experiment, our method demonstrates the state-of-the-art robustness against one of the strongest white-box attacks as PGD training, but with much less computational cost. Finally, we visualize the learning process of the network with SQA to handle strong adversaries, which is different from existing methods.

## 1 Introduction

As Convolutional Neural Networks (CNNs) stand out as a solution to many real world computer vision tasks (LeCun et al., 2015; Angelova et al., 2015; Levine et al., 2016; Litjens et al., 2017), achieving a certain level of robustness has become indispensable for security-sensitive systems, such as autonomous driving, robot vision, and identity authentication. However, recent studies (Szegedy et al., 2014; Goodfellow et al., 2015) have shown that the existing CNNs are vulnerable to small perturbations of the input that are intentionally or adversarially designed to fool the system. The adversarial attack is a serious problem since these maliciously designed attacks have shown effective in physical world scenarios, where inputs are obtained from signals of cameras and other sensors (Kurakin et al., 2016; Evtimov et al., 2017). Another disconcerting feature about adversarial examples is their transferability across different models (Szegedy et al., 2014; Papernot et al., 2016; Liu et al., 2016) that enables black-box attacks. In other words, adversarial examples can be designed from a different model without having the information about the target network.

The most studied defense strategy against adversarial attacks is adversarial training (Goodfellow et al., 2015; Kurakin et al., 2017; Tramr et al., 2018; Madry et al., 2018), which increases robustness by augmenting training data with adversarial examples. Since adversarial training requires the model to train adversarial examples in addition to training data, the model consumes extra time to learn features of the examples via fine-tuning. Even though the model is trained on more examples, it still might be defenseless to new examples generated by different attack due to the overfitting problem. Recently, Madry et al. (2018) have found that adversarial training on examples created via gradient descent with random restarts, Projected Gradient Descent (PGD) training, results in a universally and partially unbreakable model on MNIST and CIFAR-10. This method shows the state-of-the-art performance on MNIST and CIFAR-10 to the best of our knowledge, but the examples are created iteratively and the time increases proportionally to the number of steps. For instance, in our CIFAR-10 training, FGSM training on ResNet18 took less than 2 hours for 30 epochs; however, PGD training took about 30 hours for the same epochs. Thus, it is essential to find the universal method that is resistant against all of the attacks, with less computational cost.

Since high dimensional representations of the neural networks give extreme complexity to the boundary of trained manifolds (Tanay & Griffin, 2016; Dube, 2018), we start from the idea that is to reduce degrees of freedom available to the adversary. In this sense, we propose a Stochastic Quantized Activation (SQA) that provides stochastic randomness to the output of an original activation and reduces the opportunity for the attacker to make adversaries. The best advantage of SQA is that SQA with fast adversarial training, training with only FGSM examples, allows the model to have robustness comparable to PGD training with less computational cost. In particular, although SQA is one of the obfuscated gradients defined by Athalye et al. (2018), iterative optimization-based methods does not successfully circumvent our defense. Besides, SQA can be combined with any deep learning models with a few lines of code but guarantees a certain level of robustness against adversarial attacks.

In this paper, we first explain existing methods for adversarial attacks and defenses we refer in Section 2. We separate the existing defense strategies into two categories and analyze the strengths and weaknesses. In Section 3, we introduce the procedure of SQA, with an algorithm described in 1. In Section 4, we show our experimental results on MNIST and CIFAR-10 and compare with existing defense systems. Lastly, we visualize the penultimate layer of our networks and compare how SQA with fast adversarial training, learns differently from the existing methods. Section 5 concludes the work and contributions of this paper are as follows:

- We propose a Stochastic Quantized Activation (SQA) which achieves a significant level of robustness combined with FGSM training, comparable to state-of-the-art PGD adversarial training with much less computational cost.
- Due to the efficiency and the flexibility of the proposed method, it can be fastly and widely applied to any existing deep neural networks and combine with other types of defense strategies.
- We analytically demonstrate how SQA makes the model robust against adversaries in high-level and low-level by using t-SNE, and plotting activation maps.

## 2 RELATED WORK

In this section, we investigate the existing methods of adversarial attacks and defenses that appear in the following subsections. First, we define the adversarial examples with the notations formally used in this paper. Let $x$ denote input and $y$ denote the prediction of the input from the DNN classifier $f$, $y = f(x)$. Then, an adversarial example is crafted by adding a malicious noise $\eta$ into the original input $x$, causing a different prediction from the true label, $y^*$. The formal representation is as follows, where $x'$ is an adversarial example and $\epsilon$ is the noise level.

$$x' = x + \epsilon \cdot \eta, \ \text{ where } \ f(x') \neq y^* \tag{1}$$

### 2.1 GENERATING ADVERSARIAL EXAMPLES

**Fast Gradient Sign Method (FGSM)** is a fast single-step method to create adversarial examples proposed by Goodfellow et al. (2015). The authors suggest the adversarial examples are crafted because of the effects of the linear summation in DNNs, and the algorithm is as follows.

$$x' = x + \epsilon \cdot \text{sign}(\nabla_x J(f(x), y^*)) \tag{2}$$

Here $J(f(x), y^*)$ is the loss between the output prediction $f(x)$ and the true label $y^*$. However, calculating the loss based on the difference between predictions and true labels makes the label leaking effect (Kurakin et al., 2017), so one simple way to prevent it is to put the prediction $y$ instead of $y^*$. The intuition behind of the Equation 2 is that increasing loss $J$ by perturbing the input $x$ adding the gradient of loss, which makes the prediction get out of the extrema.

**Projected Gradient Descent (PGD)** is one of the strongest known white box attacks (Madry et al., 2018). It is a multi-step variant of FGSM, which means that it finds the adversarial perturbation $\eta_n$ by using the same equation from FGSM, but iteratively. What makes this attack stronger is that

it finds the adversary from starts with random $\epsilon$-uniform perturbation clipped in the range of the normalized pixel values, [0,1].

$$x_0' = Clip_x(x + uniform(-\epsilon, \epsilon)), \quad x_{n+1}' = Clip_{x,\epsilon}(x_n' + \alpha \cdot \text{sign}(\nabla_x J(f(x_n'), y^*))) \quad (3)$$

**Carlini & Wagner Attack (C & W Attack)** is strong optimization-based iterative attack proposed by Carlini & Wagner (2017). It uses Adam (Kingma & Ba, 2014) to optimize over the adversarial perturbation $\eta_n$ using an auxiliary variable $\omega_n$ and solves the equation below.

$$\begin{aligned} \text{minimize } ||\eta_n||_p \; &+ \; c \cdot f(x_n + \eta_n) \\ \text{where } \eta_n = \frac{1}{2}(&\tanh(\omega_n) + 1) - x_n. \end{aligned} \quad (4)$$

The function $f(\cdot)$ is defined as

$$f(x) = max(Z(x)_{y^*} - \max_{i \neq y^*}(Z(x)_i), -\kappa), \quad (5)$$

and we can determine the confidence with which the misclassification occurs by adjusting $\kappa$.

## 2.2 DEFENSIVE STRATEGY

**Adversarial training** increases robustness by augmenting training data in relation to adversarial examples. Previous studies (Goodfellow et al., 2015; Kurakin et al., 2017; Tramr et al., 2018) have shown that adversarially training models improve the classification accuracy when presenting them with adversarial examples. However, the intrinsic problem of this method is the high cost associated with additionally generating adversarial examples and patching them into a training batch. For this reason, practical adversarial training on a large scale dataset such as ImageNet uses fast-generated adversarial examples using FGSM only for training data. However, Madry et al. (2018) have shown that FGSM adversaries don't increase robustness especially for large $\epsilon$ since the network overfits to these adversarial examples. They instead, suggest to train the network with a multi-step $FGSM^k$, PGD adversaries, and it shows the state-of-the-art performance on MNIST and CIFAR-10.

**Obfuscated Gradients** make the network hard to generate adversaries by not having useful gradients. Recently, Athalye et al. (2018) defined three types of obfuscated gradients: Shattered Gradients, Stochastic Gradients, and Exploding & Vanishing Gradients. (Dhillon et al., 2018; Buckman et al., 2018; Song et al., 2018; Xie et al., 2018) have considered one of these gradients, but Athalye et al. (2018) make the attacks which successfully circumvent the defense by making 0% accuracy on 6 out of 7 defenses at ICLR2018. SQA can be considered as both shattered gradients and stochastic gradients. However, we found that our method does not overfit to the adversarial examples and shows robustness against the different type of attacks including the one used to break obfuscated gradients. The next section explains the details of our method.

## 3 STOCHASTIC QUANTIZED ACTIVATION

---

**Algorithm 1** Stochastic Quantized Activation

1: **function** FORWARD($h^i, \lambda$)
2:      $g^i \leftarrow (h^i - \min_{\forall j \subseteq J} h_J^i) / (\max_{\forall j \subseteq J} h_J^i - \min_{\forall j \subseteq J} h_J^i) * \lambda$
3:      $g^i \leftarrow \lfloor g^i \rceil + Bernoulli(g^i - \lfloor g^i \rfloor)$
4:      $g^i \leftarrow (g^i / \lambda) * (\max_{\forall j \subseteq J} h_J^i - \min_{\forall j \subseteq J} h_J^i) + \min_{\forall j \subseteq J} h_J^i$
5:      **return** $g^i$

6: **function** BACKWARD($\partial g^i / \partial h^i$)
7:      **return** $\partial g^i / \partial h^i$

---

In this section, we introduce the concept of SQA starting from a typical low-bit representation in DNNs as prerequisites (Courbariaux et al., 2015). Then, we show the procedure of our quantization stochasticity. The difference between typical low-bit DNNs (Hubara et al., 2016a; Courbariaux et al.,

2015; Hubara et al., 2016b) and our proposed method is that we only consider the quantization of activations except weight vectors. We found that this does not significantly slow down the training with PyTorch (Paszke et al., 2017) but maintains full-precision weight representation, which enables easier convergence than BNNs without additional training strategies.

**BinaryConnect** constraints the weights to either +1 or -1 during propagations (Courbariaux et al., 2015). Two types of binarization, deterministic and stochastic, are introduced. They are respectively described by the following equations.

$$w_b = \begin{cases} +1 & \text{if } w \geq 0, \\ -1 & \text{otherwise.} \end{cases} \tag{6}$$

$$w_b = \begin{cases} +1 & \text{with probability } p = \sigma(w), \\ -1 & \text{with probability } 1 - p. \end{cases} \tag{7}$$

$$\text{where } \sigma(x) = \text{clip}(\frac{x+1}{2}, 0, 1) = \max(0, \min(1, \frac{x+1}{2}))$$

BNNs are originally designed to reduce the significant amount of memory consumption and costs taken by propagating in full-precision networks. Recently, however, Galloway et al. (2018) shows another benefit of low-precision neural networks, which improves robustness against some adversarial attacks.

Thus, we propose SQA, a stochastic activation function giving the quantized threshold effects into vanilla CNNs, which is described in Algorithm 1. The algorithm can be divided into three steps.

- Min-Max normalization with scaling
- Stochastic Quantization
- Inverse Min-Max normalization after rescaling

Let $h^i$ be a latent space, the output from a $i$th convolutional layer after ReLU activation. We first perform min-max normalization, making $h^i$ ranging from 0 to 1. Then we scale the $h^i$ ranging from 0 to $\lambda$ by multiplying a scale factor $\lambda$, which determines the level of quantization from binary to quaternary in our experiment. In the next step, we stochastically quantize the scaled $g^i$ as $\overline{g^i}$ presented in the below equation.

$$\overline{g^i} = \lfloor g^i \rfloor + Bernoulli(g^i - \lfloor g^i \rfloor) \tag{8}$$

This makes $g^i$ converge into the closest or second closest integers, either $\lfloor g^i \rfloor$ or $\lfloor g^i \rfloor + 1$ with a probability of each, $1 - (g^i - \lfloor g^i \rfloor)$ and $g^i - \lfloor g^i \rfloor$. For instance, if we let $g^i = 1.7$, then the probability of $\overline{g^i} = 1$ is 0.3 and $\overline{g^i} = 2$ is 0.7. The final step is rescaling $\overline{g^i}$ into the range of original output ReLU activation $h^i$. To rescale the value within the original range, $\overline{g^i}$ is first divided by $\lambda$, and inverse min-max normalization is applied as presented in Algorithm 1.

Since it is impossible to find exact derivatives with respect of discretized activations, an alternative is to approximate it by a straight through estimator (Bengio et al., 2013). The concept of a straight through estimator is fixing the incoming gradients to a threshold function equal to its outgoing gradients, ignoring the derivative of the threshold function itself. This is the reason why we rescale $\overline{g^i}$ to the original range of $h^i$. In other words, we do not want to consider the scale factors multiplied in the activation function when we use a straight through estimator.

## 4 EXPERIMENT

### 4.1 DATASET AND IMPLEMENTATION DETAILS

In this experiment, we show the feasibility of our approach with several different settings on MNIST and CIFAR-10 using PyTorch (Paszke et al., 2017). We use Adversarial Box (Wang & Gavin Ding, 2018) to generate FGSM and PGD adversaries and implement C.W adversaries ($l_\infty$) based on Athalye et al. (2018). The results for each MNIST and CIFAR-10 are shown in Sec 4.2 and 4.3.

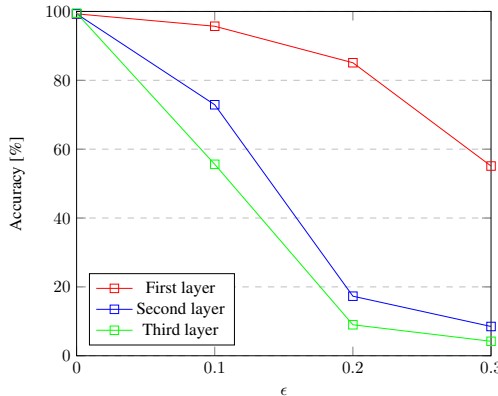

Figure 1: FGSM attack to deterministic quantization on different layers of the baseline CNN.

**Model Parameters**

For MNIST, we use a baseline model as a Vanilla CNN consisting of three convolutional layers with two fully-connected layers on the top. Since there is a correlation between robustness and model capacity (Madry et al., 2018), we use two networks with different channel sizes and increase channels by a factor of 2. This result in networks with each (16, 32, 64) and (64, 128, 256) filters and they are denoted as SMALL$and$LARGE in Table 1. We apply SQA on the first and second layers with each $\lambda = 1$ and 2. We use Stochastic Gradient Descent (SGD) with learning rate of 0.1, momentum of 0.9, and weight decay of 5e-4. We adjust the learning rate decreasing by 0.1 after every 30 steps within total 100 epochs.

For CIFAR-10, we use ResNet model (He et al., 2015) as a baseline. We adpot 34 and 101 layers of ResNets denoted as RES34 and RES101 in Table 2. An interesting property found on training CIFAR-10 dataset is that quantization with stochasticity shows much higher accuracy rather than deterministic quantization. It seems reasonable since stochasticity is able to provide higher capacity to learn complex RGB images. We apply SQA on the output from the first layer of ResNet and its bottleneck module with each $\lambda = 1$ and 2. The same hyper-parameters are applied to the MNIST training except with total epochs of 350 and decreasing learning rate by 0.1 after every 150 steps.

**Attack Parameters**

Throughout the experiments, different $l_\infty$ intensity levels are applied to the attacks. For MNIST, $\epsilon = 0.2$ and 0.3 are used for FGSM and C&W attacks to give strong adversarial perturbations. We choose 40 steps for C&W attacks. Also, we set $\epsilon = 0.2$, a step size of 0.01 and 40 steps for PGD attacks. For CIFAR-10, $\epsilon = 4, 8$ are considered for the adversarial attacks. We choose 30 steps for C&W attacks. For PGD attacks we fix 7 steps and the step size as 2 with random perturbation 8. Note that the values for MNIST are in the scale of (0,1) and (0,255) for CIFAR-10. Step sizes for the attacks are chosen to be consistent with Madry et al. (2018).

## 4.2 ATTACK ON MNIST

**Quantization on Different Layers**

Since quantizing the weights or activation lowers the accuracy on clean images (Courbariaux et al., 2015; Hubara et al., 2016a), it is important to find where to put SQA modules in networks. Thus, we investigate the $n$th layer-wise quantization applying the deterministic quantization from the first layer of CNN to the third. The result is shown in Figure 1. It is clear that applying quantization on the earlier steps gives higher robustness. This observation is another proof for the argument from Liao et al. (2017) that a small perturbation in an image is amplified to a large perturbation in a higher-level representation so that quantizing the activations in lower-level representation gives more robustness. We further, empirically found that giving binary quantization on the first layer and ternary quantization on the second layer provides less degradation for accuracy and a fair amount of robustness.

| Model | Clean | $FGSM_{\epsilon=0.2}$ | $FGSM_{\epsilon=0.3}$ | $C\&W_{\epsilon=0.2}$ | $C\&W_{\epsilon=0.3}$ | PGD |
|---|---|---|---|---|---|---|
| $SMALL_{full}$ | **99.40** | 98.48 | **98.86** | 19.43 | 0.71 | 0.50 |
| $SMALL_{SQA}$ | 96.61±0.28 | 89.30±0.19 | 82.67±0.21 | 82.39±0.17 | 57.7±0.40 | 7.43±0.26 |
| $LARGE_{full}$ | 99.25 | **98.58** | 98.21 | 64.21 | 19.49 | 0.86 |
| $LARGE_{SQA}$ | 98.64±0.34 | 95.30±0.06 | 91.14±0.24 | **92.75±0.34** | **83.62±0.33** | **83.90±0.32** |

Table 1: Performance comparison for MNIST between full-precision and SQA

| Model | Clean | $FGSM_{\epsilon=4}$ | $FGSM_{\epsilon=8}$ | $C\&W_{\epsilon=4}$ | $C\&W_{\epsilon=8}$ | PGD |
|---|---|---|---|---|---|---|
| $RES34_{full}$ | 87.07 | 46.21 | 63.47 | 13.5 | 8.20 | 8.87 |
| $RES34_{SQA}$ | 81.53±0.38 | 70.51±0.49 | 53.45±0.44 | 66.45±0.45 | 38.70±0.62 | 48.32±0.19 |
| $RES101_{full}$ | **87.67** | 52.44 | **78.59** | 12.17 | 8.09 | 7.52 |
| $RES101_{SQA}$ | 82.94±0.42 | **73.30±0.39** | 56.54±0.52 | **68.40±0.69** | **41.52±0.41** | **52.12±0.63** |

Table 2: Performance comparison for CIFAR10 between full-precision and SQA

**SQA *v.s.* Full-Precision**
We explore the robustness of SQA against three types of adversarial attacks and the result is shown in Table 1. The networks are all trained with fast single-step adversaries and we could find two known, but interesting properties from the experiments. First, FGSM training the full-precision networks, denoted as $SMALL_{full}$, $LARGE_{full}$, makes themselves overfit to the adversaries. They show depressed accuracy on especially, PGD attacks, nearly close to 0. However, SQA models does not overfit to the adversaries. Even though SQA models show lower performance on FGSM attacks, they exhibit remarkably high accuracy on the other adversarial examples that have not seen before. The second interesting fact is that the correlation between robustness and model capacity. Madry et al. (2018) have shown that increasing model capacity helps to train the network against strong adversaries successfully. Our experiment also confirms this phenomenon. The performance of $LARGE_{SQA}$ is stronger than $SMALL_{SQA}$ against FGSM attacks and more than ten times robust against PGD attacks. This result shows that the model capacity not only increases robustness against the adversaries that have been learned but also prevent overfitting to them.

### 4.3 ATTACK ON CIFAR10

**SQA *v.s.* Full-Precision**
We performed experiments on CIFAR-10 to show the effectiveness of SQA on the RGB image dataset. We tried the same types of white-box attacks as in MNIST experiments, and the result is shown in Table 2. Instead of training Vanilla networks, we adopt ResNet (He et al., 2015) since the Vanilla networks are hard to learn useful features on CIFAR-10. Two different ResNets are used for comparing robustness regarding the model capacity, and we found the same phenomena as in MNIST experiments. In other words, SQA module helps to get out of overfitting to the FGSM adversaries, and the larger capacity provides, the higher robustness against different types of attacks.

**SQA *v.s.* Other Existing Methods**
We compare our module, SQA, with recently proposed defenses including the state-of-the-art, Madry et al. (2018). We also include SAP, PixelDefend, and Thermometer (Dhillon et al., 2018; Song et al., 2018; Buckman et al., 2018) since they use stochastic gradients or shattered gradients that are one of the obfuscated gradients, where our method belongs to. Table 3 shows the performance [1] comparison against PGD and C&W attacks for $l_{\infty}(\epsilon = 8)$. Note that the architectures from the defenses on Table 3 are all different and it is impossible to exactly compare the robustness. We denote the architectures as $RESN_{W,C}^k$, where $W$ stands for Wider ResNets, $N$ is depth, $C$ is the channel size of the first layer, and $k$ is the widen factor. As Athalye et al. (2018) claimed, our method is more robust against gradient-based PGD rather than optimization-based C&W, pushing the state-of-the-art accuracy to $52\%$ against PGD attacks. Also, it shows a fair amount of accuracy against C&W attacks comparable to Adv. Training. This result shows a dramatic impact in a sense that other methods based on obfuscated gradients almost fail to defend against these strong adversaries.

---

[1] The performance of SAP, PixelDefend, and Thermometer is from Athalye et al. (2018).

| Defense | Model | PGD | C&W$_{\epsilon=8}$ |
|---|---|---|---|
| SAP (Dhillon et al., 2018) | RES20 | - | 0 |
| PixelDefend (Song et al., 2018) | RES62 | 9 | - |
| Thermometer (Buckman et al., 2018) | RES30$_W^4$ | 30 | - |
| Adv. Training (Madry et al., 2018) | RES$_W^{10}$ | 50 | **46.8** |
| SQA (Ours) | RES101 | **52.1** | 41.5 |

Table 3: Performance comparison with other defenses against $l_\infty(\epsilon = 8)$ adversaries for CIFAR10

| Defense | Training time |
|---|---|
| FGSM Training | 0.43 |
| SQA + FGSM Training | 0.57 |
| PGD Training$_{\kappa=10/20/100}$ | 1.23 / 2.34 / 10.17 |

Table 4: Average training time (sec) for one iteration on ResNet34 for CIFAR-10

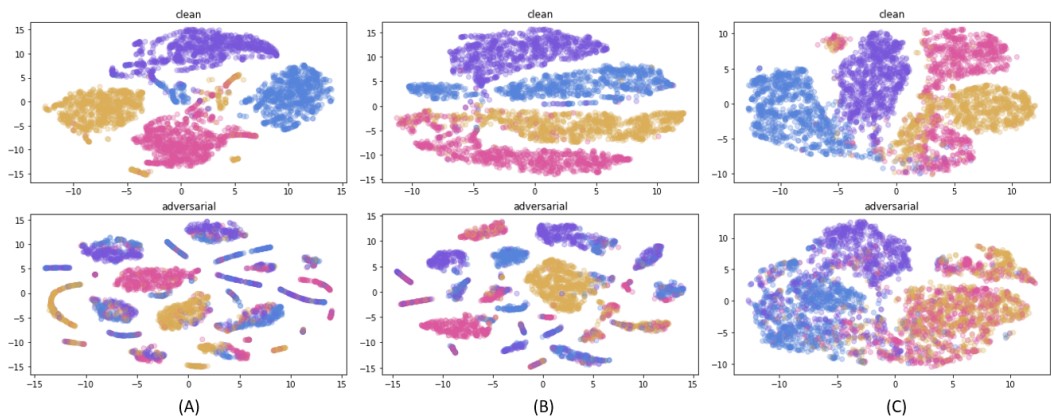

Figure 2: t-SNE results from the penultimate layer of our network against clean images and C&W adversaries. (A) is a full-precision network with no defense, (B) is a full-precision network + FGSM training, and (C) is SQA + FGSM training (Ours). While (A) and (B) shows significant degradation on adversaries, (C) still finds proper decision boundaries against adversarial attacks.

## 4.4 TIME COMPLEXITY FOR ADVERSARIAL TRAINING

In this subsection, we explore the time complexity of adversarial training both single-steps and multi-steps. Let $\tau$ as the time taken by forward and backward propagation in neural networks, $\kappa$ is number of steps to find adversaries, and $\upsilon$ is for other processing times including data loading, weight update and etc. Then, we can define the time complexity of adversarial training as follows,

$$\mathrm{T}_{Adv.Training} = (1 + \kappa) \cdot \tau + \upsilon \tag{9}$$

Then when we consider $\alpha$ as processing time for SQA module and compare SQA + FGSM training with PGD training,

$$\frac{(\kappa - 1)}{2} \cdot \tau >> \alpha \tag{10}$$

As we can see in Table 4, SQA + FGSM training is almost 18 times faster than PGD training where $\kappa$ is 100.

## 4.5 VISUALIZING PENULTIMATE LAYERS

In this subsection, we analyze the penultimate layers of the network trained with our method comparing with two full-precision networks: with no defense and with FGSM training. We use C&W

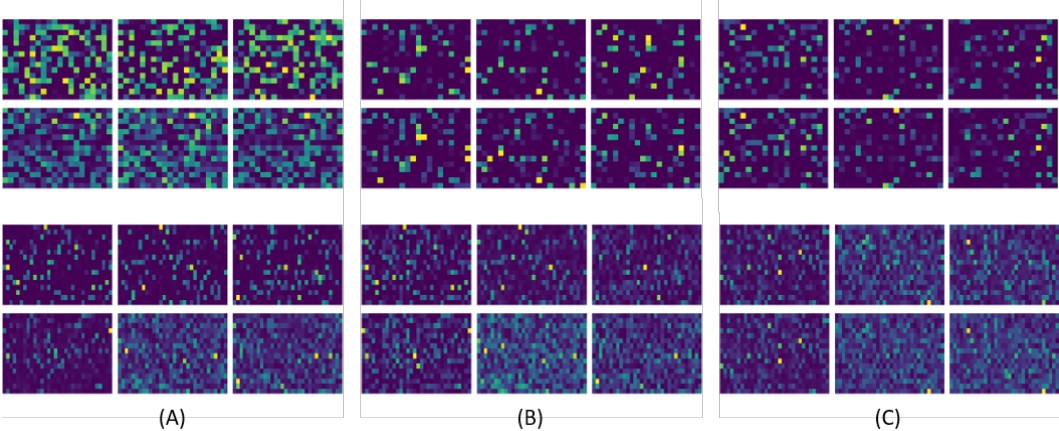

Figure 3: Activation maps from the penultimate layer of our network. Top two rows are the results on MNIST, and the bottom rows are for CIFAR-10. For each dataset, top rows are activations given inputs as clean images, and bottom rows are the ones given adversarial images. (A) is a full-precision network with no defense, (B) is a full-precision network + FGSM training, and (C) is SQA + FGSM training (Ours). C&W attacks are applied and (C) shows little difference between clean and adversaries.

attacks to make adversaries with the parameters described in Section 4.1. We use two different ways to visualize the penultimate layers in high level and low level by using t-SNE (van der Maaten & Hinton, 2008) and plotting activation maps with both clean images and adversarial examples. Firstly, Figure 2 shows t-SNE results from the penultimate layer of our network and a point in t-SNE is represented as an image. We select four classes to clearly show how the networks learn and what happens when adversarial noise is added. Here, we demonstrate that the full-precision network trained with FGSM does not correctly classify the classes against the adversarial attacks, as depicted in (B). However, only (C) which is our method shows that the clusters are less broken compared to the other methods. Furthermore, in light of the fact that the robust classifier requires a more complicated decision boundary (Madry et al., 2018), our model seems to have the complicated one by learning adversarial examples.

Secondly, we closely look into the penultimate layer in a low level by plotting the each of the activations. In this time, a point of an activation map stands for the mean value of the activations across about a thousand images per classes. We found that the yellow spots which are the highest values stay in the same location under the adversarial attack, as depicted in (C), Figure 3. In other words, our method shows stable activation frequencies against the adversarial attacks, but training full-precision models with FGSM adversaries does not help to increase robustness, as shown in (B).

## 5 CONCLUSION

In this paper, we have found that SQA, a stochastic quantization in an activation function, make existing neural networks prevent overfitting to FGSM training. It provides stochastic randomness in quantization to learn a robust decision boundary against adversarial attacks with FGSM training. Our method not only shows dramatic improvements against one of the strongest white-box attacks, comparable to state-of-the-art PGD training but also significantly reduces the computational cost. Throughout visualizing the penultimate layers of our network, we demonstrate that the network learns strong adversaries without overfitting. We expect that SQA could be fastly and widely applied to other defense strategies because of its efficiency and flexibility. In the future work, we plan to experiment on large scale image datasets.

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
