# OpenReview forum: "Stochastic Quantized Activation: To prevent Overfitting in Fast Adversarial Training"
_ICLR.cc/2019/Conference_

### Official Review · AnonReviewer2 · 2018-11-01
**Interesting work but requires more thorough experiment**

**Rating:** 4
**Confidence:** 4

**Review:**

This paper proposes to use a  stochastically quantized network combined with adversarial training to improve the robustness of models against adversarial examples. The main finding is that, compared to a full precision network, the quantized network can generalize to unseen adversarial attacks better while training only on FGSM-perturbed input. This provides a modest speedup over traditional adversarial training.

While the findings are certainly interesting, the method lacks experimental validation in certain aspects. The comparison with other adversarial training methods is not standardized across networks, making the efficiency claims questionable. Furthermore, I am uncertain whether the authors implemented expectation over transformations (EoT) for the C&W attack.  Since the network produces randomized output, vanilla gradient descent against an adversarial loss is likely to fail. It is conceivable that by taking an average over gradients from different quantizations, the C&W adversary would be able to circumvent the defense better. I would be willing to reconsider my review if the authors can address the above weaknesses.

Pros:
- Surprising result showing that quantization leads to improved generalization to unseen attack methods.

Cons:
- Invalid comparison to other adversarial training techniques since the evaluated models are very different.
- Lack of evaluation against EoT adversary.
- Algorithm 1 is poorly presented. I'm sure there are better ways of expressing such a simple quantization scheme.
- Figures 2 and 3 are uninteresting. The fact that the model is robust against adversaries implies that the activations remain unchanged when presented with perturbed input.

---

### Official Review · AnonReviewer1 · 2018-11-04
**Limited novelty, but good experimental results**

**Rating:** 5
**Confidence:** 5

**Review:**

The paper proposes to quantize activation outputs in FGSM training. The algorithm itself is not novel. The straight through approach for training quantized network has been used in previous papers, as also pointed out by the authors. The new thing is that the authors found that quantization of activation function improves robustness, and the approach can be naturally combined with FGSM adversarial training. Experimental results show comparable (and slightly worse) results compared to adversarial training with PGD, while the proposed approach is faster in training time.

I have the following questions/comments:

1. Why not do SQA with PGD-adversarial training? If SQA+FGSM performs similar to PGD training, SQA+PGD might perform even better.

2. There are several important papers missing in the discussion/comparisons:
- Quantization improves robustness has been reported in a previous paper: "Defend Deep Neural Networks Against Adversarial Examples via Fixed andDynamic Quantized Activation Functions". How does the proposed algorithm compare with this paper?
- Adding stochastic noise in each layer has been used in some recent papers: "Towards Robust Neural Networks via Random Self-ensemble". It will be good to include into discussions.

3.  I can't find the comparison between PGD-training and SQA on MNIST. Are they also comparable on MNIST? Showing results on more datasets will make the conclusion more convincing.  If the benefit of the proposed approach is training time, showing the scalability on ImageNet will make the argument stronger.

---

### Official Review · AnonReviewer3 · 2018-11-05
**interesting idea, but too much of an accuracy hit, and a problem with clarity**

**Rating:** 4
**Confidence:** 3

**Review:**

The paper proposes a model to improve adversarial training, by introducing random perturbations in the activations of one of the hidden layers. Experiments show that robustness to attacks can be improved, but seemingly at a significant cost to accuracy on non-adversarial input.

I have not spent significant time on adversarial training, and review the paper under the following understanding: It was observed that the decision regions of a class are sprinkled with "holes" that get misclassified. These holes are neither naturally occuring. Their existence allows a potential attacker to coerce a model into mis-classifying by providing specially crafted inputs, in order to attain a benefit. Therefore, those holes are called "adversarial" examples. The risk is heightened by the fact that adversarial examples are commonly not mis-classified by humans (or even detectable by the eye). To "plug" the holes, one includes adversarial examples in the training, called "adversarial training." A resulting system should now have a much improved accuracy for the "holes", while ideally not affecting classification accuracy for the natural examples, which will continue to constitute nearly 100% of the samples the system will be used on. (The "hole" metaphor may not be entirely appropriate, since the space of adversarial examples that are neither misclassified by humans nor detectable is likely much larger than the space of naturally occuring samples.)

The paper proposes a way of plugging the hole by quantizing layer activations. The results show that this makes the system robust to adversarial attacks.

Clarity:

I spent a lot of time figuring out, as someone who has not spent a lot of time with this, what is being evaluated. It is very unclear whether the non-clean systems in Tables 1 and 2 do apply FGSM etc. also in training (in combination with SQA), or only to the test samples. Table 4, the wording in 4.2, and the wording of the Conclusion indicate that they are. But then, where do I find the accuracy on the naturally-occuring (non-manipulated) samples?

The only combination of interpretations that makes sense in the end is to parse "The networks are all trained with fast single-step adversaries" as to mean "The networks are all trained with FGSM", and that the non-Clean columns in Table 1 refer to test data perturbed by the respective method, while the Clean column shows the accuracy on the natural data. This *must* be clarified in the final version, as it took way too long to understand this. I strongly suggest to do this with the naming: change small_full to small_FGSM, and small_SQA to small_SQA+FGSM.

Assuming I figured this out right, the tables still lack the baseline accuracy of doing nothing (clean-clean), so one can know how much the nearly-100% use case gets affected.

Results:

The second concern I have is that, assuming my reading of the results as described above is correct, that the SQA method quite severely affects accuracy on the clean test data, e.g. increasing the error rate on CIFAR by 72% (from 12.33% to 17.06%). There must be a discussion on why such severe performance hit is worth it, especially since there often is an accuracy cliff below which there is a steep loss of usability of a system. For example, according to my personal experience in speech recognition, the difference between 12% and 17% is the difference between decent and unacceptable user experience (also considering that a few percent of errors are caused by ambiguities in the ground-truth annotations themselves, which should be the case for CIFAR as well).

Figure 1 seems a little misleading in this regard since the areas of good accuracy are very condensed. It should be rescaled, as only the area close to the optimum performance is relevant. It does not matter whether we degrade from 99.x% to 77% or 58%, or even 95-ish. All of those hurt performance to the point of not being useful.

It would be nice to discuss what an accuracy metric would be that is useful for the end user. It would have to be a combination of the expected cost of a misclassification of a natural image and the expected cost caused by attacks. A good method would improve this overall metric. A paper attempting to address adversarial attacks should at least discuss this topic briefly, in my view.

Technical soundness:

A technical question I have is whether the min-max normalization may be too susceptible to outliers. A single extreme activation can drastically shift the threshold for \lambda=1. How about a mean-var normalization? If there is batch or layer normalization in the system, your activations may already be scaled into a consistent range anyway, that might allow you to use a constant scaling on top of that.

Another question I have is: quantization is often modeled as adding uniform noise. Why not add noise directly? And why uniform noise? For example, would compute g = h + Gaussian noise with std dev=(max-min)/lambda work equally well? What is special about quantization?

And another technical question: My guess is that the notable loss of accuracy is caused by the strong quantization (two values only in the case of \lambda=1). I think the paper should show results for larger lambdas, specifically whether there is a better trade-off point between the accuracy loss from quantization vs. robustness to adversarial samples.

Section 3/SQA: "This is the reason why we rescale g^i to the original range of h^i" This seems wrong. I think the main reason is that one would not want to totally change the dynamic ranges of the network, as it may affect convergence merely by scaling. You'd want to limit any impact on convergence to the quantization itself.

Significance:

I think the significance is limited. Given that the accuracy impact of the mitigation method is very large, I do not consider this paper as substantially solving the problem, or even bringing a practical solution much closer in reach.

Pros:
 - tnteresting idea;
 - comparison against various attacks.

Cons:
 - Hard to understand because it was left unclear what is evaluated, at least to readers who are not familiar with a possibly existing implied convention;
 - The method seems to harm accuracy on clean data a lot, which is the main use case of such a system.

I would in the current form reject the paper. To make it acceptable, the clarity of presentation, especially of the results, must be improved, but more importantly, more work seems necessary to reduce the currently significant accuracy hit from the method, and the trade-off of quantization level vs. robustness should be addressed.

Minor feedback:

Please review the paper for grammar and spelling errors (e.g. "BinaryConnect constraints" or the use of "make", which is often not correct).

In Algorithm 1, I suggest to not use 'g', as it may be mis-read as "gradient." Unless this is a common symbol in this context.

"Thus, we propose SQA" warrants another \subsubsection{}, to indicate where \subsubsection{BinaryConnect} ends.

Section 2.2's early reference to SQA is a little confusing, since SQA has not formally been defined. I would smooth this a little, e.g. change "SQA can be considered" to "We will see that our SQA, as introduced in the next section, can be considered"

"an alternative is to approximate it" probably should be "our approach is to approximate it"

---

### Meta-Review · Area_Chair1 · 2018-12-17
**reject**

**Confidence:** 4
**Recommendation:** Reject

**Metareview:**

While the paper contains interesting ideas, the reviewers suggest improving the clarity and experimental study of the paper. The work holds promises but is not ready for publication at ICLR.